# Gelsolin, an Actin-Binding Protein: Bioinformatic Analysis and Functional Significance in Urothelial Bladder Carcinoma

**DOI:** 10.3390/ijms242115763

**Published:** 2023-10-30

**Authors:** Abeer A. Alsofyani, Taoufik Nedjadi

**Affiliations:** King Abdullah International Medical Research Center, King Saud Bin Abdulaziz University for Health Sciences, Jeddah 21423, Saudi Arabia; alsofyaniab@ngha.med.sa

**Keywords:** bladder cancer, gelsolin, biological functions, prognosis, TCGA, immune cell infiltration

## Abstract

The involvement of the actin-regulatory protein, gelsolin (GSN), in neoplastic transformation has been reported in different cancers including bladder cancer. However, the exact mechanism by which GSN influences bladder cancer development is not well understood. Here, we sought to reveal the functional significance of GSN in bladder cancer by undertaking a comprehensive bioinformatic analysis of TCGA datasets and through the assessment of multiple biological functions. GSN expression was knocked down in bladder cancer cell lines with two siRNA isoforms targeting GSN. Proliferation, migration, cell cycle and apoptosis assays were carried out. GSN expression, enrichment analysis, protein–protein interaction and immune infiltration analysis were verified through online TCGA tools. The data indicated that GSN expression is associated with bladder cancer proliferation, migration and enhanced cell apoptosis through regulation of NF-κB expression. GSN expression correlated with various inflammatory cells and may influence the immunity of the tumor microenvironment. Computational analysis identified several interacting partners which are associated with cancer progression and patient outcome. The present results demonstrate that GSN plays an important role in bladder cancer pathogenesis and may serve as a potential biomarker and therapeutic target for cancer therapy.

## 1. Introduction

Bladder cancer (BLCA) is one of the most commonly diagnosed neoplasms in the urinary system, and it is associated with high mortality rates. An updated report released by GLOBOCAN in 2020 approximated that 573,278 new cases were diagnosed and that 212,536 patients died of bladder cancer worldwide. Interestingly, the incidence of BLCA is steadily rising across the globe representing 3% of all cancers in 2020 [1]. Many risk factors have been associated with BLCA pathogenesis, including age, gender, obesity, heavy exposure to occupational chemicals (such as aromatic amines), frequent tobacco smoking and genetic predisposition [1].

Non-muscle invasive bladder cancer (NMIBC) accounts for around 70–80% of all cases, most of which reach better prognosis if diagnosed at an early stage [2]. However, NMIBC showed high rates of recurrence within 5 years and high risk of progression to aggressive muscle-invasive bladder cancer (MIBC) [3]. Patients with MIBC are subjected to a stringent treatment plan requiring either cystectomy or radical radiotherapy plus/minus chemotherapy, resulting in poor prognosis with the 5-year survival rate less than 50% [4].

This situation necessitates the search for biomarkers for early diagnosis of bladder cancer, as early detection ensures better management and good prognosis for bladder cancer patients. Several useful markers are being used in the clinic to assist in the diagnosis, prognosis and monitoring of disease progression, including the grading and the TNM staging systems. Nonetheless, there is a lack of definitive and comprehensive clinical markers that would help with cancer prognostic, risk stratification and proper management of cancer patients [2,3]. Furthermore, there is a pressing need for novel biomarker-driven treatment strategies in bladder cancer. We have recently utilized proteomic-based approaches to identify non-invasive biomarkers for early detection of bladder cancer using blood-derived plasma samples [5,6,7,8]. Using combined 2DE-DIGE and mass spectrometry, our findings identified a number of differentially expressed proteins that could serve as potential biomarkers associated with muscle-invasive phenotype (MIBC). One of the proteins that we discovered to be significantly downregulated in cancer patients compared to healthy controls was gelsolin (GSN) [9].

GSN is an actin-capping protein mainly found in the cytoplasm, plasma and mitochondria. It promotes cell motility by regulating actin filaments’ assembly and disassembly [10]. In addition, GSN is involved in many cellular functions such as modulating cell morphology, proliferation and apoptosis [11,12]. GSN plays an important role in controlling many signal transduction pathways. Altered expression of GSN was associated with many different diseases including cancer [11]. Increased GSN expression level has been reported in breast [13], prostate [14], hepatocellular carcinoma [15] and pancreatic cancers [16] where it acts as an oncogene. Conversely, GSN was categorized as a tumor suppressor in several other malignancies, in which GSN expression was reduced including, colon [17], non-small-cell lung [18], ovarian [19], gastric [20] and urinary bladder cancers [21]. Interestingly, a decline in the plasma GSN (pGSN) level has been implicated in various illnesses including infection, trauma and inflammation [22]. Circulating pGSN has been suggested as a novel diagnostic biomarker since its level is significantly declined in many cancer types. It has been demonstrated that both circulating and cytoplasmic GSN expression were associated with patient outcome [23,24,25]. However, the exact mechanism by which GSN influences bladder cancer development is not well understood. Here, we sought to investigate the functional significance and the underlying mechanism of GSN in bladder cancer.

## 2. Results

### 2.1. Pan-Cancer Analysis of GSN mRNA Expression

In our previous study, we reported a decrease in GSN expression in bladder cancer using proteomic analysis [9]. Here, we used The Cancer Genome Atlas (TCGA) dataset to evaluate the expression pattern of GSN in various types of tumors and matching them with normal tissues by applying the GEPIA portal. As shown in Figure 1a, GSN expression was not consistent throughout the cancer sites. GSN expression was significantly lower in breast carcinoma, cervical carcinoma, colorectal adenocarcinoma and uterine carcinoma compared with the corresponding normal tissues. On the other hand, B-cell lymphoma, GBM, acute myeloid leukemia glioma, hepatocellular carcinoma, prostate adenocarcinoma and thymoma exhibited significantly high levels of GSN compared to the expression levels in normal tissues. The results from the TCGA dataset revealed that GSN mRNA expression was significantly lower (*p* < 0.01) in cancerous bladder tissues compared to that in adjacent non-cancerous tissues (Figure 1b). These data are consistent with our previous finding [9] confirming the downregulation of GSN expression in bladder cancer. Therefore, we opted to comprehensively explore the functional significance of GSN expression in bladder cancer.

### 2.2. GSN Expression in Bladder Cancer Lines

In order to develop a comprehensive overview of the role of GSN in bladder cancer, a functional characterization was carried out. We first assessed the expression level of GSN in a panel of multiple bladder cancer cell lines including 5637, EJ, T24 and SW780 using Western blotting analysis. Our data indicated that, with the exception of SW780, all three cell lines 5637, EJ and T24 showed high expression levels of GSN (Figure 2a). Therefore, the T24 cell line was selected for use in all subsequent experimental assays, as it exhibited higher expression of GSN than other cell lines and also due to its high growth ratio (Figure 2b). Next, T24 cells were transfected with two siRNA isoforms targeting GSN (siGSN#1 and siGSN#2). Two non-targeted siRNA sequences (siNC#1 and siNC#2) were used as controls. Total proteins were extracted after transfection, and GSN protein expression was analyzed using Western blot. The results indicated that siRNA-mediated GSN depletion significantly impaired the protein level in T24 cells (Figure 2c). The siRNA knockdown reduces the GSN protein expression by more than 80% (Figure 1d). Interestingly, siGSN#1 was more potent in inhibiting GSN expression compared to siGSN#2; therefore, all subsequent assays were performed using siGSN#1. siRNA-mediated GSN knockdown was further validated in T24 cells using immunofluorescence microscopy. Figure 2e showed that the staining intensity is significantly lower in siRNA-transfected GNS cells compared to siRNA-transfected control cells. To further understand the impact of GSN on bladder cancer progression, a series of in vitro assays were designed to explore the functional significance and the underlying mechanisms of GSN expression in bladder cancer.

### 2.3. Downregulation of GSN Reduces Cell Proliferation

To explore the potential role of reducing GSN expression on T24 cell proliferation, an MTT assay was performed using siGSN#1 and two control siRNA-transfected T24 cells. The data revealed an absorption at 450 nm wavelength (optical density–OD), which reflects the growth potential of the cells. The absorbance values of the siGSN#1-treated cell were expressed as a ratio of that in siRNA-treated control cells. The reading was recorded at 24 h, 48 h and 72 h post-transfection (Figure 3a). siGSN#2 only modestly reduced the proliferation rate of T24 compared to control siRNAs, which might be associated with the potential of this isoform to knock down GSN level. Our data demonstrated that OD in siGSN#1 treated cells was significantly lower compared with siRNA-treated control cells (*p* < 0.001), indicating that depletion of GSN is accompanied by a reduction in the proliferation potential of bladder cancer cells.

### 2.4. siRNA-Mediated GSN Knockdown Suppresses Cell Migration

We next questioned whether reducing GSN levels may influence the migration of bladder cancer cells. Seventy-two hours after transfection with GSN and control siRNAs, T24 cells were grown for an extra 24 h in 24-well plates to form a confluent monolayer. Wound healing assay showed that GSN knockdown significantly (*p* < 0.001) impairs the migration capacity of the T24 cells by 84% (Figure 3b,c). Similar to the proliferation assay, siGSN#2 did not impair cell motility of T24.

### 2.5. The Effects of GSN Knockdown on Cell Cycle

Flow cytometry using propidium iodide staining was employed to investigate the effect of GSN knockdown on the cell-cycle progression at 72 h post-transfection. Our data indicated no difference in the cell-cycle phases between siGSN#1-transfected cells and siRNA-transfected control T24 cells (*p* < 0.231), suggesting that GSN is not involved in the cell-cycle progression of T24 cell line (Figure 4a,b).

### 2.6. GSN Knockdown Induces Apoptosis of Bladder Cancer Cells

Then, we sought to evaluate the cellular apoptotic status of T24 cells following siRNA depletion of GSN using an annexin V-FITC flow cytometry assay. As shown in Figure 4c,d, 72 h post-transfection with GSN-targeting siRNA#1, the proportion of apoptotic T24 cells was significantly increased in the GSN knockdown group compared with siRNA-treated and untreated control groups (*p* < 0.004). Furthermore, we explored the effects of siRNA-mediated GSN knockdown on caspase 3/7 activity. The results revealed that the caspase-3/7 activity of cells transfected with siGSN#1 was increased compared with control siRNA-transfected cells (*p* < 0.001, Figure 4e). This result indicates that GSN depletion induces the cells to undergo apoptosis. Taken together, these data suggest that inhibiting GSN expression decreases cell viability by triggering apoptosis of bladder cancer cells. 

### 2.7. GSN-Downregulated Expression Suppresses the NF-κB Pathway

In order to investigate the downstream signaling pathways that mediate bladder cancer in GSN knockdown, we examined the effect of GSN downregulation on NF-κB p65 expression. It is well documented that NF-κB has the capacity to coordinate the transcription of genes involved in cell proliferation and apoptosis. After 72 h of transfection, Western blotting showed that NF-κB p65 and phosphorylated NF-κB p65 protein levels were significantly reduced in response to decreased GSN expression compared to control groups (Figure 5a,b). These results suggest that downregulation of GSN may influence bladder cancer development through alteration of NF-κB expression.

Next, we monitored the NF-κB translocation under GSN siRNA and control siRNA-treated conditions with immunofluorescence. GSN knockdown was repeated in T24 bladder cancer cells and confirmed with immunofluorescence (Figure 6a). The obtained data are consistent with Figure 1a,b indicating that knockdown of GSN impairs NF-κB expression compared to control siRNA-treated cells. Furthermore, no NF-κB translocation is seen following GSN knockdown (Figure 6a).

### 2.8. Relationship between GSN Expression Level and Clinicopathological Characteristics

To gain an in-depth understanding of the involvement of GSN in bladder carcinogenesis, TCGA dataset analysis was performed using the UALCAN platform to assess the association between GSN gene expression level and a number of patients’ clinical and pathological parameters. As reported earlier, TCGA data indicated significant suppression of GSN expression in bladder cancer tissues compared to non-cancerous tissues (Figure 1b). We then explored the possible mechanism of GSN gene silencing by assessing gene methylation status. It is well perceived that DNA promoter methylation is a common mechanism for epigenetic regulation of gene expression. Our results indicate that reduced GSN mRNA expression is highly attributed to the binding of methyl groups on the GSN promoter since data analysis demonstrated that the promoter methylation level of GSN in BLCA was significantly higher compared to normal tissues (*p* < 0.01, Figure 7a). Additionally, we further analyzed the association between GSN expression level and bladder cancer molecular subtypes (neuronal, basal squamous, luminal, luminal infiltrated and luminal papillary) and found that the GSN expression level was downregulated in all subtypes compared to their normal counterparts (*p* < 0.01, Figure 7c).

Moreover, the results revealed that down-expression of GSN is an early event in bladder cancer development and is significantly correlated with high tumor stage (stages 2, 3 and 4) of BLCA patients (*p <* 0.01, Figure 7b). Interestingly, as shown in Figure 7d, patients with high expression levels of GSN had significantly poorer overall survival than those with low expression levels (*p* = 0.008). These findings suggest that GSN may serve as a prognostic biomarker and may be a potential candidate for targeted therapy.

### 2.9. Analysis of GSN Gene Co-Expression

Further analyses were undertaken to predict the potential molecular mechanisms by which GSN prompts tumorigenesis in BLCA. Here, we screened for GSN’s target and co-expressing gene networks through the UALCAN dataset. Gene ontology (GO) analysis was conducted, and hundreds of co-expressed genes were identified, some of which were positively co-expressed with GSN expression and others negatively co-expressed with GSN.

The heatmaps in Figure 8a,b showed the top twenty-five genes that positively and negatively correlated with GSN expression. Among the genes that are positively enriched with GSN, we found MYL9, LIMS2, FLNA, CALD1, TGFB1I1, TAGLN, RGS2, TPM2, PALLD and KANK. Concomitantly within the list of genes that showed a strong negative relationship with GSN expression, we identified the following: NDUFA7, TIMM16, NSUN6, IP6K2, COX7C, GATA3, FAM174B, MGST2, KRTCAP3 and JMJD7-PLA2G4B. 

### 2.10. Functional Enrichment Analyses of GSN

To predict the functional enrichment information of GSN inter-activating genes, gene ontology (GO) analysis was performed using Metascape database. Identified GSN-related genes participated in the regulation of many cellular processes including the positive and negative regulation of the biological process, the metabolic process, the immune system process, cell signaling, development and growth (Figure 8c).

We next used the Molecular Signatures Database (MSigDB Hallmark 2020) through Enrichr software (https://maayanlab.cloud/Enrichr/) to characterize the pathway involvement of the positively GSN associated gene set. Based on the functional enrichment analysis, the positively regulated gene set is mainly involved in the following hallmarks: epithelial–mesenchymal transition, inflammatory response, myogenesis, TNF-alpha signaling via NF-kB, interferon gamma response, coagulation, apical junction, IL-2/STATS signaling and allograft rejection (Figure 8d).

### 2.11. Protein–Protein Interaction Network Analysis

The protein–protein interaction (PPI) network of genes interacting with GSN was further validated using the GeneMANIA platform. The PPI network of interacting genes with GSN consisted of 21 nodes including CASP9/3, TRAF3IP2, APAF1, ACTA1, BAX, CAPG and PIK3C2A. The functional annotation of the gene set enriched for GSN was mainly responsible for positive regulation of cysteine-type endopeptidase activity involved in the apoptotic process, apoptotic nuclear changes, positive regulation of proteolysis, positive regulation of peptidase activity, actin filament capping, actin filament depolymerization and regulation of release of cytochrome c from mitochondria (Figure 8e). 

### 2.12. Relationship between GSN and Immune Cell Infiltration

Published data indicated that immune cell infiltration could predict prognosis of bladder cancer patients [26]. Hence, the relationship between GSN expression and the immune cell infiltration profile was explored using the TIMER database. The algorithm evaluated an association between the expression of GSN and the abundance of six immune cell subtypes demonstrating that GSN expression is significantly negatively associated with tumor purity (*p* = 1.48 × 10^−24^). Moreover, significant positive correlations between GSN expression level and CD8^+^ T cells (*p* = 7.74 × 10^−5^), CD4^+^ T cells (*p* = 7.47 × 10^−4^), macrophages (*p* = 1.53 × 10^−14^), neutrophils (*p* = 6.98 × 10^−7^) and dendritic cells (*p* = 7.93 × 10^−12^) were reported. No significant correlation with infiltrating levels of B cells was identified (r = −0.075, *p* = 1.56 × 10^−1^) in bladder cancer (Figure 9).

## 3. Discussion

Bladder cancer remains one of the most challenging diseases to treat due to its recurrence and progression potentials. The search for potential biomarkers associated with early cancer development, progression and prognosis is a promising approach to tackle the disease and assist in the development of novel therapeutic strategies [17,24,27,28]. 

We have previously undertaken a systematic proteomic approach (2D-DIGE/MS) to identify novel candidate markers associated with muscle-invasive bladder cancer (MIBC) progression using patients’ plasma samples. One of the identified differentially regulated proteins was GSN. It was significantly downregulated in patients with high-grade (HG) muscle-invasive tumors compared to healthy controls [9]. Although the link between GSN expression level and cancer progression has been suggested, the molecular mechanisms governing its role are not well understood in bladder cancer. In the present study, we aimed to further characterize the functional significance of GSN and delineate its regulatory network in bladder cancer.

GSN is an actin-modulating protein expressed in the cytoplasm and extracellular fluids. As a multifunctional regulatory molecule, cytoplasmic GSN plays an important role in cytoskeletal remodeling and cell motility. It also plays a role in regulating cell apoptosis, signal transduction and transcriptional co-activation [11]. It is widely recognized that GSN has been associated with several pathological processes such as inflammation, sepsis and in predicting cancer progression [11,24,29]. Here, we performed bioinformatics analysis to examine GSN expression across several malignancies including bladder cancer using TCGA database. Our results highlight the divergence of GSN expression between different cancer types. Interestingly, this apparent dis-regulation of GSN expression has been previously reported by different studies where GSN has been suggested serve as either a tumor suppressor or tumor activator depending on the tumor type and tumor stage [30,31]. The TCGA data on GSN are consistent with our previous finding [9], suggesting that reduced GSN expression in advanced cancer (MIBC) may play an important role in bladder cancer progression. It has been shown that upregulated GSN expression contributes to cell proliferation, migration and invasion in human oral squamous cell carcinoma Tca8113 [32]. The upregulated expression of GSN was also reported in human colon cancer cells that led to an enhanced capacity for cell migration [33]. The increased motility of the cells plays an important role in tumor progression, particularly during invasion and metastasis. Published data showed that GSN played an important role in melanoma cell migration. GSN-depleted A375 melanoma cells exhibited a significantly impaired capacity to migrate on laminin, a primary component of the skin’s basement membrane [34]. Changes in cellular morphology imply alteration of cell cytoskeleton proteins, including GSN that also plays an important role in cell invasion and metastasis. Thompson et al. demonstrated that a reduction in GSN levels significantly inhibited pancreatic cancer cell motility [16]. These findings are in line with the results of the present study, which shows that the depletion of GSN expression is associated with loss of bladder cancer cell motility using the wound healing assay. Yuan et al. (2013) indicated that reduced ATF3 expression suppressed bladder cancer metastasis through GSN-mediated actin remolding [35]. Similarly, Shieh et al. (1999) support the hypothesis that high GSN expression can facilitate tumor dissemination and metastasis by promoting tumor cell locomotion, which can subsequently translate into poor clinical outcomes [36]. The investigators speculated that the involvement of GSN in cellular motility might be the underlying reason for the poor prognostic effect of high GSN expression. 

Simultaneously, we found that knocking down GSN expression reduces proliferation and increases apoptosis of bladder cancer cells in MTS assay and annexin-V assay, respectively. The proliferative potential of GSN was previously reported in colon cancer. Kim et al. (2018) demonstrated that GSN-overexpressing LoVo colon cancer cells have significantly increased proliferative pattern and possess a two-fold greater invasive potential compared to control cells [37]. These data are also consistent with our findings. Of note is that one siRNA isoform (siGSN#1) produced a significant reduction in the GSN expression level compared with siGSN#2; hence, the data presented are related to one isoform. This difference might be related to the difference in oligonucleotide sequence and the target area of the gene.

The involvement of GSN in the neoplastic transformation of bladder cancer through proliferation, migration and apoptosis was previously reported. Many studies have attempted to understand the mechanism of action of GSN-mediated apoptosis. It was postulated that increasing GSN expression exerts its anti-apoptotic effects by blocking the release of cytochrome c from mitochondria, resulting in an inhibition of a cascade of aspartate-specific cysteine proteases: caspase-3, caspase-8 and caspase-9. Another proposed mechanism was related to the fact that GSN can bind to voltage-dependent anion channels and block the mitochondrial permeability of cytochrome. In hepatocarcinoma HepG2 cells, GSN was found to repress p53-mediated apoptosis by its binding to p53 in the cytosol and inhibiting p53 translocation to the nucleus [38]. In addition, it has been shown that upregulated expression of the GSN promotes radioresistance in non-small cell lung cancer cells and inhibits radiation-induced apoptosis through activation of PI3K/Akt signaling. In head and neck cancer (HNC) cells, reducing the expression of GSN induced an apoptotic response to cisplatin [39]. Moreover, Abedini et al. (2014) demonstrated that low levels of GSN were found in chemo-sensitive patients, and overexpression of GSN is associated with an aggressive behavior of gynecological cancers [40]. These data emphasized the important role gelsolin may play in carcinogenesis. In bladder cancer, the link between GSN expression and apoptosis was reported by Nowak et al. The authors demonstrated that addition of nicotine-derived metabolites to the T24 cell line increased the expression of GSN and promoted cell proliferation, migration and apoptosis [41]. In addition, Miura et al. reported that GSN is one of the downregulated proteins in cisplatin-resistant bladder cell line HT-1376 [42]. Cisplatin is a potent inducer of apoptosis; however, published data indicated that GSN could confer resistance against chemotherapeutic agents and inhibit apoptosis by blocking the voltage-dependent anion channel [43]. Moreover, protein–protein interaction and pathway enrichment analysis revealed that GSN interacts with cytochrome c, which is known to inhibit apoptosis.

In the current study, we attempted to explore the mechanism by which GSN may induce apoptosis. The transcription factor NF-kappa B might serve as a potential candidate. There is convincing evidence that the transcription factor NF-kappa B (NF-κB) signaling pathway is associated with cancer cell proliferation, survival, migration and metastasis [44,45,46]. It has been demonstrated that suppressing the activation of the NF-κB pathway resulted in impeding cell growth and boosting apoptosis via transcription activation of a number of anti-apoptotic genes, including Bcl-xL, IEX-1L, Bfl-1and X-IAP [47,48]. In bladder cancer cells, activation of the NF-κB pathway was found to inhibit apoptosis and enhance bladder cancer cell proliferation [49]. Previous findings reported that modulating the NF-κB signaling pathway could be considered a potentially effective therapeutic strategy in platinum-resistant bladder cancer [50]. In biological network analysis, a closed protein interaction was found between GSN and the oncogene NF-κB in cervical cancer [51]. Another study showed that toll-like receptor-dependent NF-κB translocation was blocked by GSN in astrocytes as a result of septic shock from bacterial endotoxin [52]. Our present study demonstrates, for the first time, that decreased expression of GSN leads to a diminished expression of NF-κB in human bladder cancer cells, T24. Furthermore, GSN knockdown failed to induce nuclear translocation of NF-κB. Further detailed investigations are needed to understand the molecular dynamics of GSN interaction with the NF-κB pathway.

In order to comprehensively investigate the role of GSN in bladder carcinogenesis, we analyzed the publicly available TCGA data through the UALCAN portal. The results showed that GSN is significantly downregulated in bladder cancer compared with normal tissue (Figure 1a,b). Moreover, decreased expression of GSN appears to be an early event in bladder cancer development since reduced expression of GSN is noticed at stage one of cancer development and stays at low levels through all stages. This may be a safe and protective mechanism to prevent the progression of bladder cancer. This is an interesting finding because early cancer detection is of paramount important in the clinical management of cancer patients [53]. The usefulness of circulating plasma GSN in early cancer diagnosis was previously reported in ovarian patients [54]. Our finding revealed that the GSN gene is highly methylated in cancer compared to normal tissues, which explains the low expression of GSN and suggests that GSN hypermethylation may play a role in bladder cancer development. It has been demonstrated that aberrant gene silencing through promoter methylation is an early event in the development of many cancers [55]. Epigenetic downregulation of GSN was reported in breast and bladder carcinomas [56,57]. Wu et al. (2022) reported that GSN hypermethylation was associated with a poor prognosis for patients with gastric cancer [58]. These findings suggest that GSN hypermethylation could serve as a biomarker for bladder cancer development. Interestingly, we observed that GSN downregulation occurred across all molecular subtypes of bladder cancer including neuronal, luminal and basal. Molecular subtypes of bladder cancer were previously reported as key features of disease heterogeneity and is strongly considered in the clinical management of bladder cancer patients [59,60]. Furthermore, our results showed that GSN expression correlated significantly with immune cell infiltration, especially with CD4^+^, CD8^+^ T-cells as well as macrophages, neutrophils and dendritic cells. This finding reflects the importance of the immune tumor microenvironment (TME) in cancer development, progression and distant metastasis formation. A growing piece of evidence shows the role of GSN in mediating the interplay between cancer cells and TME through the remodeling of the extracellular matrix and selective suppression of immune cells [61]. There has been a report showing the correlation between GSN and various immune cells’ infiltration, especially dendritic cells in gastric cancer, and how decreased GSN expression is associated with poor patient outcome [58]. A few other reports indicated the survival benefit of GSN under-expression in osteosarcoma and lung cancer [58,62]. This is not consistent with Kim et al. (2018), who reported significantly greater 5-year DFS rates in GSN under-expression compared to the opposite group in colon cancer [37]. Taken together, these data suggest that GSN could serve as a tumor suppressor or promoter depending on tumor type. 

To obtain further insight on the molecular mechanism of GSN-mediating bladder tumorigenesis, gene ontology (GO) analysis was performed. The latter has emerged as a fundamental tool in biological studies providing knowledge on the biological processes of sub-ontology, molecular function sub-ontology and cellular component sub-ontology [63]. It is well perceived that protein functions are accomplished via interaction with other proteins in a particular process and specific location [64]. Hence, GO mapping for GSN-interacting genes and pathways may provide a resource for the interpretation of the observed functional outcome following GSN knockdown.

A heatmap of the top 25 positively and negatively regulated genes revealed that the primary biological processes involve epithelial to mesenchymal (EMT) transition, inflammatory response, myogenesis, TNF-α signaling, coagulation and apical junction. These biological processes increase tumorigenicity. EMT is believed to maintain the stemness properties of cancer cells, whereas inflammation is a critical component and a hallmark of cancer, as many cancers are triggered from inflammation sites by cytokines released and mediated via the TNF-α pathway [65,66]. Moreover, the GeneMANIA analysis of protein–protein interaction demonstrated that caspase-3, caspase-9, TRAF3IP2, APAF1, CAPG, CYCS, BAX and ACTA1 represent the primary binding proteins. An examination of the functions of the listed GSN-interacting proteins using the pathway enrichment tool demonstrated that these genes are enriched in pathways regulating apoptosis, proteolysis, peptidase activity, actin filament capping and polymerization in addition to cytochrome c regulation. The involvement of GSN-binding proteins in cancer progression is also well documented [67,68,69,70]. For instance, regulation of pro-apoptotic genes BAX, caspase-3 and caspase-9 is a major event in cancer development and progression [71]. Similarly, CapG and ACTA1, which are major components of the cell cytoskeleton, are also implicated in cancer initiation and progression and are linked to the outcome of cancer patients [72]. In conclusion, the current study indicates that GSN expression is associated with bladder cancer proliferation, migration and enhanced cell apoptosis through inhibition of the expression of NF-κB. In addition, GSN expression was correlated with various inflammatory cells and may influence the immunity of the tumor microenvironment. Computational analysis of the TCGA database identified several interacting partners which are associated with cancer progression and patient outcome. The present results demonstrate that GSN plays an important role in bladder cancer pathogenesis and that targeting GSN may serve as a potential therapeutic option for cancer therapy.

## 4. Materials and Methods

### 4.1. Cell Culture

The human bladder cancer-derived cell lines 5637, T24 and SW780 were purchased from the American Type Culture Collection (Manassas, VA, USA). The EJ cell line was a gift from Dr. T Elmitwalli (University of Liverpool, Liverpool, UK). The cells were cultured in T75 flasks at 37 °C in a humidified incubator with 5% CO_2_ atmosphere in DMEM, (Gibco) supplemented with 10% fetal bovine serum (FBS) and 1% of penicillin–streptomycin solution (100X; 10,000 IU penicillin and 10,000 µg/mL streptomycin). The cells were sub-cultured every 3 days during the logarithmic growth phase. The cells were washed 3 times with 1x phosphate-buffered saline (PBS) and trypsinized with 1x trypsin-EDTA (0.25%) solution.

### 4.2. RNA Interference and Cell Transfection

T24 cells were plated in a 6-well plate at 1.5 × 10^5^ cells per well. After 24 h, cells were washed twice with PBS and transfected with 25 nm siRNA using Lipofectamine RNAiMAX reagent (Invitrogen, Waltham, MA, USA) in accordance with manufacturer’s instructions. The following ON-TARGET plus human siRNAs against GSN: hGSN siRNA#1: J-007775-06-0020; hGSN siRNA#2: J-007775-07-0020; as well as the non-targeting siRNA controls: D-001810-01-50 and D-001810-02-50 from Dharmacon (Lafayette, CO, USA) were used. After 72 h of transfection, the cells were harvested for subsequent experiments.

### 4.3. Western Blotting Analysis

Transfected cells were lysed in RIPA buffer (89901, Thermo Fisher Scientific, Waltham, MA, USA) containing a protease inhibitor cocktail (5892970001, Roche, Basel, Switzerland). The lysates were then centrifuged at 13,000 rpm for 20 min at 4 °C, and the supernatant was stored at −20 °C. The protein concentrations were quantified using Qubit protein assay (233211, Thermo Fisher Scientific, Waltham, MA, USA). A total of 30 µg of proteins was loaded and separated on 8% SDS-PAGE. Proteins were transferred to nitrocellulose membranes, blocked with 5% skimmed milk for 1 h at room temperature, then incubated with the primary antibodies overnight at 4 °C. GSN antibody was purchased from ProteinTech (Rosemont, IL, USA), beta-actin was purchased from Abcam (Cambridge, UK). Membranes were washed three times with 1x PBST for 5 min each and incubated with IRDye-labeled secondary antibodies for 1 h at room temperature. After a series of washings (3 × 10 min), the protein bands were then visualized using an Odyssey Infrared Imaging System (LI-COR Biosciences, Lincoln, NE, USA), and densitometry analysis was carried out using ImageJ version 1.8 software (National Institute of Health, Bethesda, MD, USA). The primary antibodies used in Western blot experiments were as follows: rabbit anti-GSN polyclonal antibody (Protein tech group, 1:1000), NF-kB (Cell signaling, 1:1000) and monoclonal anti-β-actin (Sigma, Ronkonkoma, NY, USA, 1:1000). All experiments were repeated at least three times.

### 4.4. Immunofluorescence Assay

T24 cells were seeded at 1.5 × 10^5^ cells per well on sterile glass coverslips in a 6-well plate. Cells were treated with siRNAs against GSN and controls for 72 h. Cells were washed with PBS, then fixed with 4% paraformaldehyde in PBS overnight at 4 °C. After three washing cycles with ice-cold PBS, the cells were permeabilized with 1% Triton x-100 in PBS for 10 min. Cells were then washed with PBS three times for 5 min each and blocked with 1% bovine serum albumin (BSA) in PBST for 1 h. Subsequently, coverslips were incubated with GSN or NF-κB (Cell Signaling Technology, Beverly, MA, USA) primary antibodies overnight at 4 °C then probed with goat anti-rabbit IgG secondary antibody conjugated to Alexa Fluor-555 or Alexa Fluor-488 (Thermo Fisher Scientific, Waltham, MA, USA). Following three washing cycles with PBST, the coverslips were mounted using anti-fade mounting solution containing DAPI. Image acquisition was performed using a Zeiss LSM710 confocal microscope (Carl Zeiss GmbH, Jena, Germany).

### 4.5. Cell Viability Assay

The quantitative determination of cell proliferation after GSN protein knockdown was evaluated using EZ4U non-radioactive cell proliferation assay following the manufacturer’s protocol. 

After 72 h of siRNA transfection, cells were seeded at a density of 10,000 cells per well in five replicates in a 96-well plate. A total of 20 µL of substrate dye solution was added to each well, and the cells were incubated for 4 h at 37 °C. Absorbance of the formazan production was measured using a microplate reader SpectraMax 384 Plus (Molecular Devices, San Jose, CA, USA) at wavelengths of 450 nm and 620 nm for background reference. 

### 4.6. Migration Assay

Transfected cells were seeded in 24-well plates (5 × 10^5^ cell/well) and allowed to adhere for 24 h at 37 °C. A wound was created by scratching the confluent monolayer cells using sterile P-200 Pipetman tips. Images were taken at time zero and after 9 h using an EVOS inverted microscope (Life Technologies, Grand Island, NY, USA) and the wound gap area was measured using ImageJ version 1.8 software (National Institute of Health). The wound closure was calculated using the following equation = area (T_9h_ − T_0h_)/area T_0h_, (where T_9h_ = time at 9 h; T_0h_ = time zero). All experiments were independently performed in triplicate.

### 4.7. Cell Cycle Analysis

After 72 h of transfection, the cells were harvested, washed with 1x PBS and fixed with ice-cold 70% ethanol overnight at 4 °C. One day later, T24 cells were again washed with 1x PBS and stained with (2 μg/mL) propidium iodide (PI) and RNase. The cell cycle was analyzed using a FACSCanto flow cytometer (BD bioscience, San Jose, CA, USA).

### 4.8. Apoptosis

siRNA-transfected cells undergoing early/late apoptosis were analyzed with annexin V-FITC and PI staining using annexin an V-FITC assay kit (Cayman Chemical, Ann Arbor, MI, USA). The cells were collected, washed with 1x PBS, then resuspended in 200 µL of 1x binding buffer. The cells were then incubated with 50 µL of annexin V-FITC/PI solution in the dark at room temperature for 10 min, followed by the addition of 150 µL of 1x binding buffer. The cells were immediately analyzed using a FACSCanto flow cytometer (BD bioscience, San Jose, CA, USA).

### 4.9. Caspase-3/7 Activity Detection

Caspase-3/7 enzymatic activities in the GSN-impaired T24 cells and controls were measured using Apo ONE^®^ Homogeneous Caspase-3/7 Assay Kit (G7790, Promega, Madison, WI, USA), as per the manufacturer’s instructions. Enzymatic activity was determined at three different times (24 h, 48 h and 72 h) following the transfection. After 72 h of siRNA transfection, cells were seeded in a 96-well plate, at a density of 10,000 cells (100 µL) per well in five replicates and allowed to adhere overnight. An equal volume of Apo-ONE caspase-3/7 reagent was added to each well and incubated for a further 3 h at room temperature. The fluorescence of each well was measured (excitation: 485 nm/emission: 530 nm) using a Synergy HT Multi-Mode Microplate Reader (Biotech, VA, USA). Caspase-3/7 activity is expressed as ratio of fluorescence changes in GSN knockdown/untreated control × 100.

### 4.10. Bioinformatic Analyses

In order to evaluate the expression levels of GSN in distinct types of tumors, we used the Gene Expression Profiling Interactive Analysis (GEPIA) webserver that extracts data from The Cancer Genome Atlas (TCGA) and the GTEx database of normal tissues http://gepia.cancer-pku.cn/ accessed on 16 December 2022) [73]. To predict the prognostic values of GSN in BLCA, we evaluated the GSN gene expression in BLCA based on sample type, histological subtypes, molecular subtypes and individual cancer stage using the UALCAN data portal (http://ualcan.path.uab.edu/analysis.html/ accessed on 16 December 2022) [74]. We also used UALCAN to evaluate epigenetic regulation of GSN expression using promoter methylation and to obtain a list of positive and negatively correlated genes with GSN. For functional and pathway enrichment analysis of the positively correlated genes of GSN, Metascape (https://metascape.org/gp/index.html#/main/step1/ accessed on 1 February 2023) [75] and Enricher (https://maayanlab.cloud/Enrichr accessed on 1 February 2023) [76] were used for further gene annotation and analysis. To construct the protein–protein interaction (PPI) networks and predict the function of the GSN gene, we used GeneMANIA (https://genemania.org/ accessed on 1 February 2023), an online tool that can analyze gene co-expression, physical interaction, gene co-location, gene enrichment analysis and website prediction [77]. The Tumor Immune Estimation Resource (TIMER) database (http://timer.cistrome.org accessed on 16 December 2022) was used to explore the relationship between GSN gene expression and the infiltration levels of different immune cells in BLCA tissues [78]. GSN expression and immune cell abundance were correlated using Spearman’s correlation analysis. Positive or negative correlation is indicated by a coefficient of correlation > 0.3.

### 4.11. Statistical Analysis

All values were represented as mean ± SD. The statistical significance of the data was assessed using Student’s *t*-test (GraphPad Prism 9.0 software). *p* < 0.05 was considered significant.

## Figures and Tables

**Figure 1 ijms-24-15763-f001:**
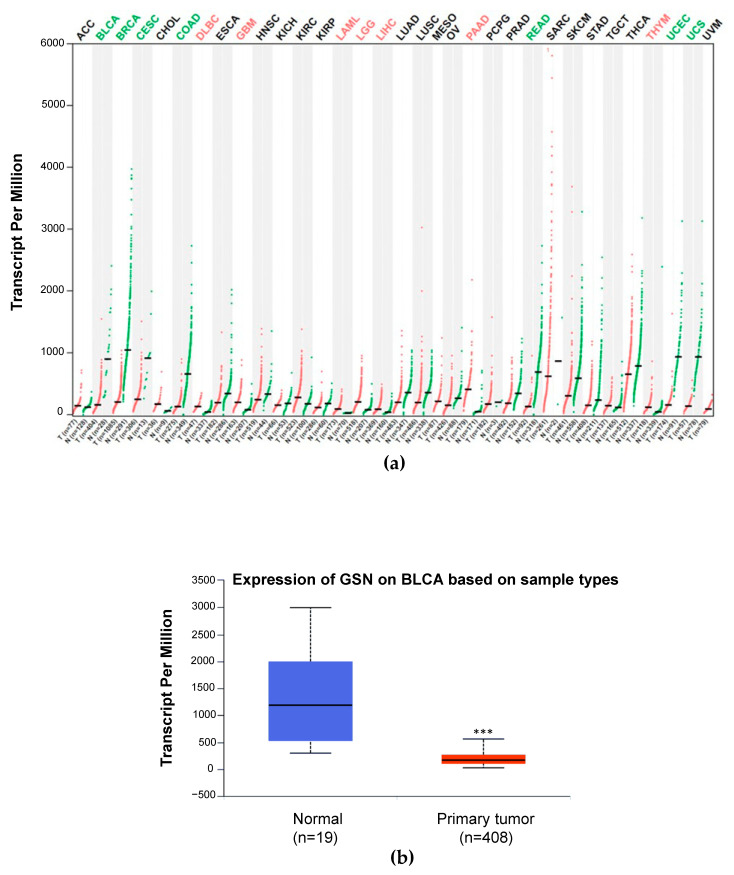
Expression of GSN in TCGA data. (**a**) Dot plot shows the expression level of GSN across tumors compared to normal tissue using GEPIA webserver. Black font represents no significant difference between tumor tissue and normal tissue. Green represents GSN expression in normal tissues (N), and red represents GSN expression in the tumor tissues (T). (**b**) The box plot shows the relative expression level of GSN in normal and BLCA tissues. ACC: adenoid cystic carcinoma; BLCA: bladder urothelial carcinoma; BRCA: invasive breast carcinoma; CESC: cervical squamous cell carcinoma; CHOL: cholangiocarcinoma; COAD: colon adenocarcinoma; DLBC: diffuse large B-cell lymphoma; ESCA: esophageal carcinoma; GBM: glioblastoma; HNSC: head and neck squamous cell carcinoma; KICH: kidney chromophobe; KIRC: Kidney renal clear cell carcinoma; KIRP: Kidney renal papillary cell carcinoma; LAML: acute myeloid leukemia; LGG: low-grade gliomas; LIHC: liver hepatocellular carcinoma; LUAD: lung adenocarcinoma; LUSC: lung squamous cell carcinoma; MESO: mesothelioma; OV: ovarian carcinoma; PAAD: pancreatic adenocarcinoma; PCPG: pheochromocytoma and paraganglioma; PRAD: prostate adenocarcinoma; READ: rectum adenocarcinoma; SARC: sarcoma; SKCM: skin cutaneous melanoma; STAD: stomach adenocarcinoma; TGCT: tenosynovial giant cell tumors; THCA: thyroid carcinoma; THYM: thymoma; UCEC: uterine corpus endometrial carcinoma; UVM: uveal melanoma. *** *p* < 0.001.

**Figure 2 ijms-24-15763-f002:**
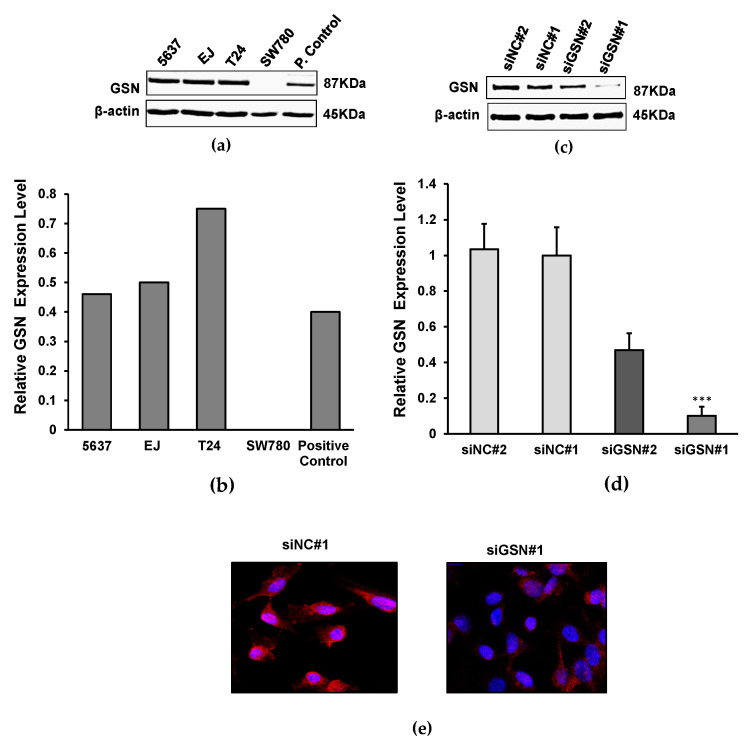
Expression of GSN in bladder cancer cell lines. (**a**) Evaluation of GSN protein expression in different bladder cancer cell lines (5637, EJ, T24 and SW780) using Western blotting analysis. Cell lysates were separated on SDS-PAGE and immunoblotted with anti-GSN antibody. β-actin expression was used as an endogenous control. (**b**) Bar chart representing densitometric fold change in GSN protein in all cell lines normalized to β-actin expression. (**c**) Western blot analysis of the levels of GSN proteins and β-actin in T24 cells after 72 h transfection with siRNAs specific for GSN and two non-targeted controls. Cell lysates were separated on SDS-PAGE and immunoblotted with anti-GSN and anti-β-actin antibodies. β-actin expression was used as an endogenous control. (**d**) The diagram shows the relative GSN fold change normalizing to β-actin expression levels following 72 h transfection with siRNAs and control with non-targeted siRNA. siGSN#1 significantly reduced the protein expression of GSN (*** *p* < 0.001) compared to siGSN#2. (**e**) Validation of GSN-targeted knockdown of T24 cells transfected with GSN-siRNAs and non-targeted control siRNA (red) with immunofluorescence staining. Nuclei were stained with DAPI (blue). All data are represented as the mean ± SD of three independent experiments. *** *p* < 0.001.

**Figure 3 ijms-24-15763-f003:**
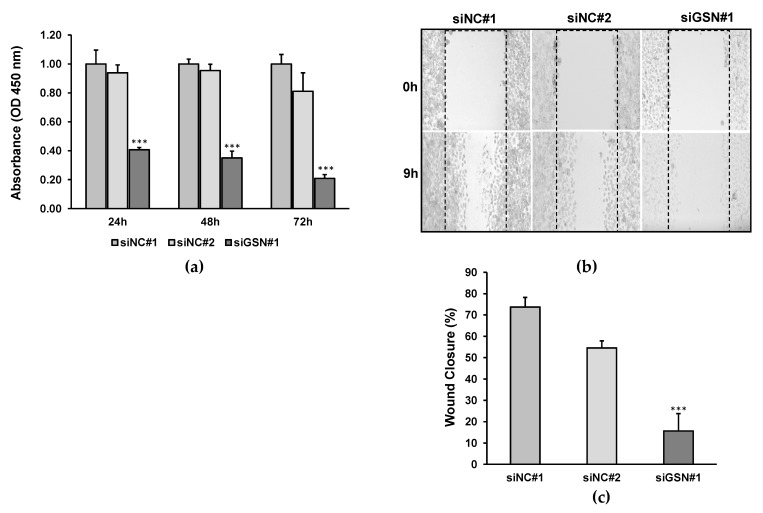
Knocking down the GSN expression inhibited the proliferation and migration of T24 cells. (**a**) MTT assay was performed 72 h after transfection with GSN-targeting and control siRNAs. Histogram showing the absorption at 450 nm wavelength in siGSN#1-treated cells was significantly lower compared with siRNA-treated control cells. (**b**) Depletion of GSN impairs T24 cell wound closure. Wound healing assay was performed 72 h after transfection with GSN-targeting and control siRNAs. (**c**) Histogram illustrates the motility index for wound healing assay using T24 cells. All data are represented as the mean ± SD of three independent experiments. *** *p* < 0.001.

**Figure 4 ijms-24-15763-f004:**
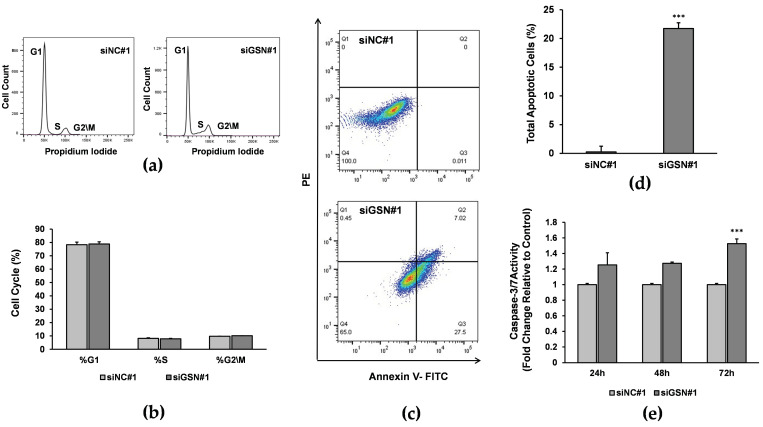
Effect of GSN knockdown on cell cycle and apoptosis. (**a**) Cell cycle distribution of T24 cells 72 h post-transfection with GSN-specific siRNA (siGSN#1) or non-targeted control siRNA (siNC#1). (**b**) Bar chart represents the percentage of cells in G1, S and G2/M phases of the cell cycle following GSN knockdown. The cell number in each phase was calculated using FlowJo software (https://www.flowjo.com/). (**c**,**d**) Annexin V-FITC flow cytometry shows a significant increase in the apoptosis of T24 cells after 72 h of transfection with GSN siRNA (siGSN#1) compared with nontargeted control treated siRNA group (siNC#1). (**e**) Caspase-3/7 activity of T24 cells transfected with siGSN#1 was increased compared with the caspase-3/7 activity of cells transfected with control siNC#1 as determined by an Apo-ONE Homogeneous Caspase-3/7 assay. All data are represented as the mean ± SD of three independent experiments. *** *p* < 0.001.

**Figure 5 ijms-24-15763-f005:**
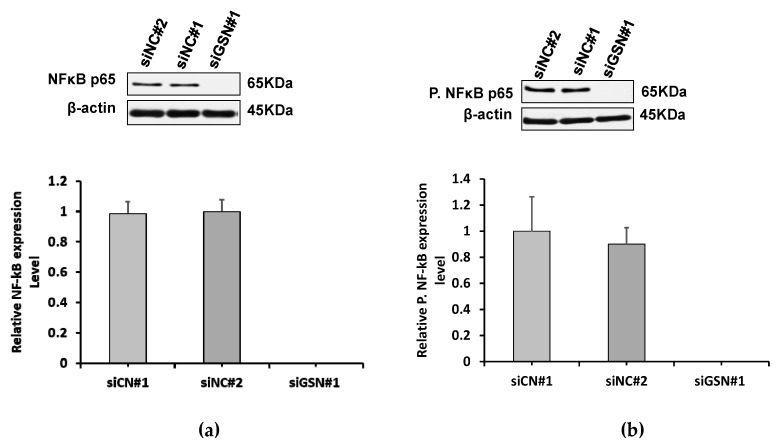
Effect of GSN downregulation on NF-kB expression. Seventy two hours after GSN transfection with GSN (siGSN#1) and control siRNAs (siNC#1 and siNC#2), T24 cell lysates were probed with antibodies against NF-kB p65 and anti-phospho-NF-kB p65 antibody. Reducing the expression of GSN is associated with inhibition of (**a**) NF-κB and (**b**) phospho-NF-κB in T24 cells compared with siRNA-treated control groups. Housekeeping β-actin expression was used as an endogenous loading control for Western blotting. Bar charts represent the relative expression of the densitometric bands related to NF-kB p65 and phospho-NF-kB p65 expression levels. All data are represented as the mean ± SD of three independent experiments.

**Figure 6 ijms-24-15763-f006:**
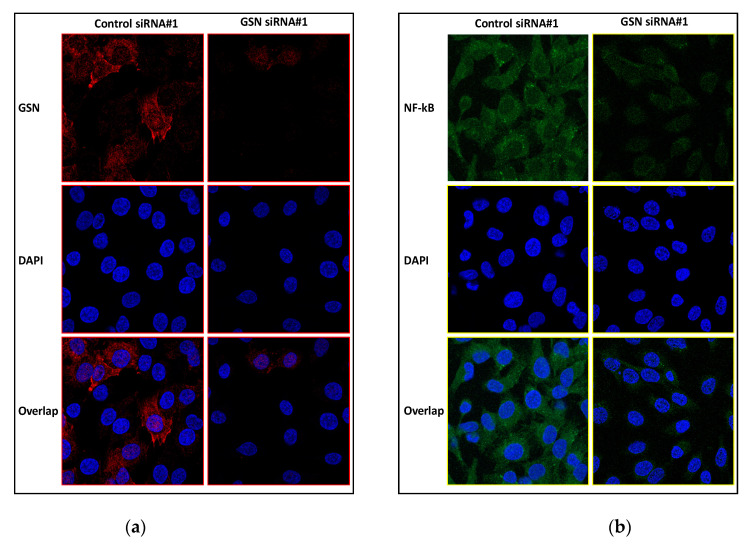
GSN downregulation has no effect on NF-kB nuclear translocation. Immunofluorescence (IF) of T24 cells for the detection of GSN or NF-kB performed 72 h after transfection with GSN#1 siRNA or control#1 and #2 siRNAs. (**a**) GSN knockdown in T24 cell was confirmed by IF using GSN-tagged Alexa-555 antibody (red dye). (**b**) Cytoplasmic and nuclear expression and translocation of NF-kB following GSN knockdown in T24 cell line using NF-kB-tagged Alexa-488 antibody (green dye). Nuclei were stained with DAPI (blue dye).

**Figure 7 ijms-24-15763-f007:**
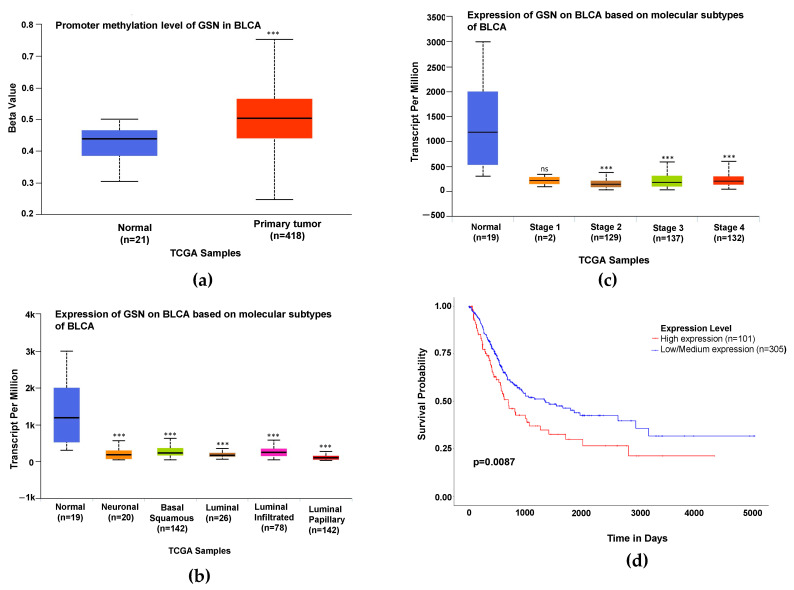
Relationships between GSN mRNA expression level and clinicopathological characteristics of bladder cancer patients. (**a**) Promoter methylation levels of GSN in bladder cancer. Box whisker plot showing high promoter DNA methylation level of GSN in normal and BLCA tissues. (**b**) Expression of GSN in BLCA based on cancer stages (stages 1–4). (**c**) Correlation between GSN expression and BLCA molecular subtypes, i.e., neuronal, basal squamous, luminal, luminal infiltrated and luminal papillary, respectively. (**d**) Overall survival analysis of bladder cancer patients based on GSN expression. Kaplan–Meier overall survival indicating that high expression level of GSN was positivity correlated with poor overall survival in BLCA patients compared to patients with low GSN expression. The UALCAN database of TCGA was used to generate the survival plot. All data are represented as the mean ± SD. ns: not significant. *** *p* < 0.001.

**Figure 8 ijms-24-15763-f008:**
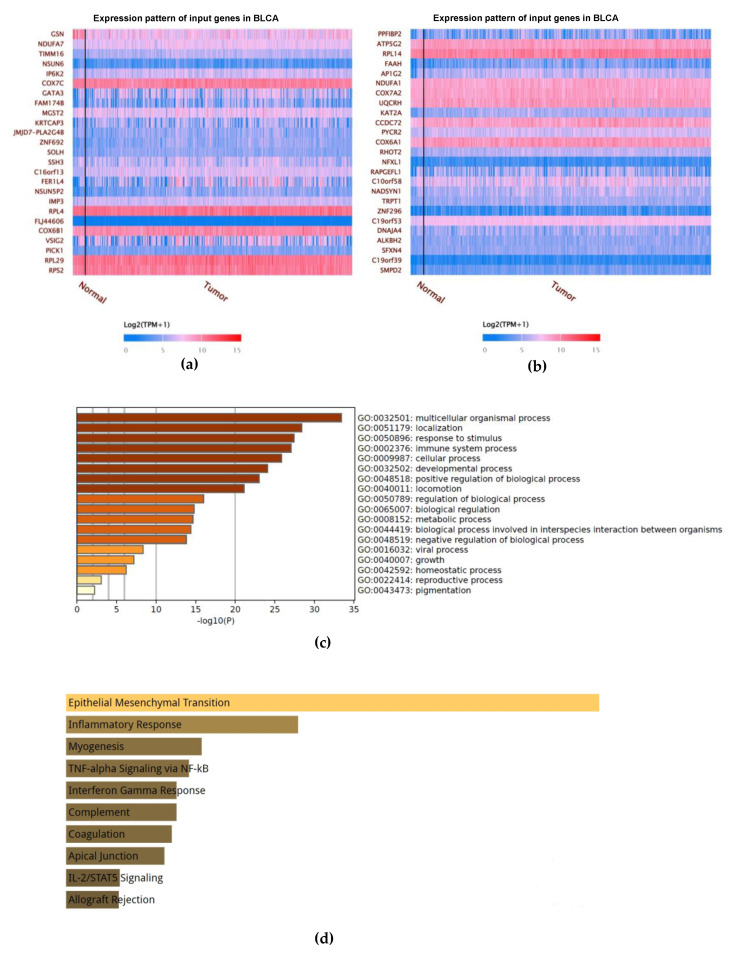
Functional enrichment analysis of GSN in BLCA. (**a**) Heatmap representing top 25 genes positively correlated with GSN expression in BLCA. (**b**) Heatmap representing top 25 genes negatively correlated with GSN in BLCA. The red means high expression of genes, and blue means low expression of genes. (**c**) Gene ontology (GO) analysis across positively GSN associated gene lists. Orange is the enrichment term and is colored by *p*-values. (**d**) Bar charts showing significant terms for MSigDB Hallmark 2020 following analysis of the positively GSN associated gene set, ranked by *p*-value. (**e**) Protein–protein interaction (PPI) network of GSN was constructed via the GeneMANIA website. PPI network and functional analysis indicating the gene sets that were enriched in the target network of GSN. The interactions between proteins in distinct colors of the network edge indicate the bioinformatic methods applied: physical interactions, co-expression, predicted, co-localization, pathway, genetic interactions and shared protein domains. The size of the nodes reflects the node degree in the network—the higher the degree, the bigger the node. The distinct colors for the network nodes indicate the biological functions of the sets of enrichment genes.

**Figure 9 ijms-24-15763-f009:**
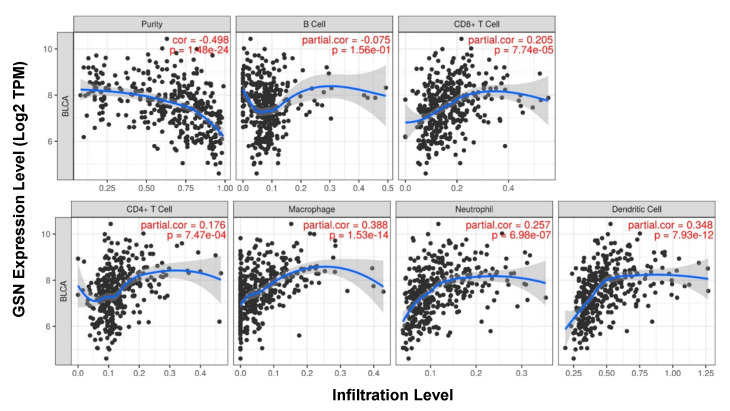
Immune cell infiltration analysis in bladder cancer. Correlation between the expression level of GSN and the abundance level of six immune cells. The data demonstrated a strong association between GSN expression and B cells, CD8^+^ T cells, CD4^+^ T cells, macrophages, neutrophils and dendritic cell infiltration in BLCA as analyzed using the TIMER database.

## Data Availability

The datasets used and/or analyzed during the current study are available from the corresponding author on reasonable request.

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
