# Peer review of "Gelsolin, an Actin-Binding Protein: Bioinformatic Analysis and Functional Significance in Urothelial Bladder Carcinoma"

_ijms, 2023, doi:10.3390/ijms242115763_

Round 1
Reviewer 1 Report
The article by Alsofyani and Nedjadi aims to investigate the functional significance of the actin-regulatory protein gelsolin (GSN) in bladder cancer development. In a precedent work of Nedjadi and coworkers published in 2021 (https://doi.org/10.3390/life11121294) based on a proteomics-based approach, GSN was found downregulated in the plasma of bladder cancer patients with respect to healthy controls. In the present article, the authors analyze the role of GSN in bladder cancer by means of functional assays in cell culture integrated by bioinformatic analysis.
In the Introduction the authors report that GSN has an ambivalent behavior in different cancers, acting as an oncogene in some types of cancers (such as breast, prostate, hepatocellular, and pancreatic cancers) and as antioncogene in others, including bladder cancer. Ovarian cancer is reported in both cancer lists and it would be interesting a comment from the authors on that.
The bioinformatics analysis of the TGCA dataset reported in the present article confirms an opposite trend in GSN expression in different types of cancers. In particular, the analysis of bladder cancer samples shows a downregulation in bladder cancerous tissues compared to adjacent non-cancerous tissues (Figure 1B), in perfect agreement with previous data from Nedjadi and coworkers (2021) obtained in plasma. These data indicate then GSN can act as a suppressor of bladder cancer. Therefore, it is quite surprising that all the experiments have been made downregulating GSN expression in a T24 bladder cancer cell line, which has a high GSN expression. Surprisingly GSN downregulation seems to suppress cancer features. How these data can be reconciled to previous expression data in patients and to the other bioinformatics data produced in the article and how can they help to dissect the putative tumor suppressor role of GSN in vivo?
I believe that additional experiments, such as GSN downregulation in normal urothelial cell lines and GSN overexpression in bladder cancer cell lines, such as SW780, according to the data presented here, would be fundamental to clarify GSN role in a wider context.
A comparison of different features (i.e. proliferation and migration rate) and GSN expression levels in different bladder cancer cell lines, compared to normal urothelial cells, might be also informative.
Reviewer 2 Report
1. The X-axis is not clear in Fig 1A.
2. The position of siNC cells is too close to the edge of quarter 1. The parameters may be adjusted for accurate measurement.
3. Sections 2.5 and 2.6 may be combined.
4. The authors should present p65’s nuclear translocation in Fig 5.
5. The authors mention that this study’s purpose is to uncover the molecular role of GSN in BLCA tumorigenesis. Thus more data may be provided for the relationship between GSN and apoptosis.
6. Many genes’ expression levels are the same between normal and tumor, e.g., COX7C, IMP3, RPL4, RPL29, and RPS2 in Fig 7A.
7. More interpretation may be required for the data in Fig. 7, although some interpretation is mentioned in the Discussion.
8. Why are the results in Fig 8 included in this study? It seems that the data in Fig 8 are not related to the molecular role of GSN.
9. All the x-axis legends are not evident in Fig 6.
Moderate editing is required.
Round 2
Reviewer 1 Report
From the literature and the expression data presented in the article, it seems that GSN can act as a tumor suppressor in bladder cancer, as the authors themselves agree in their reply.
As raised in my previous report, it is unusual to study a putative tumor suppressor gene by downregulating its expression in a tumor cell line (T24), that does show a good GSN expression level. Even more unexpectedly, GSN downregulation in T24 cell line causes the reduction of cancer features, such as cell proliferation and migration, and apoptosis increase. The data obtained in this single cell line would point instead to an oncogenic role of GSN in bladder cancer. Therefore, to try to dissect this relevant issue, I suggested additional experiments, such as:
- GSN overexpression in another bladder cancer cell line, such as SW780, that as authors report do not express GSN and then might recapitulate better bladder cancer behavior in vivo
- GSN downregulation in normal bladder cells
- Functional assay in other bladder cancer cell lines, besides T24, and correlation of the obtained results with GSN expression levels.
However, the authors did not try to perform any of these experiments, producing no new data. The issue that authors rise on the long time they would need to get normal bladder cancer cells from ATCC is not a sufficient justification to avoid any type of experiment, even the ones with the bladder cancer cell lines they already have. Therefore, I am really sorry, but at this point, I cannot recommend this article for publication.
Author Response
The authors would like to thank the reviewer for the insightful and constructive comments, which help improving the quality of our manuscript. All responses to the comments are provided below in italics.
Reviewer#1
- GSN overexpression in another bladder cancer cell line, such as SW780, that as authors report do not express GSN and then might recapitulate better bladder cancer behavior in vivo.
- GSN downregulation in normal bladder cells
- Functional assay in other bladder cancer cell lines, besides T24, and correlation of the obtained results with GSN expression levels.
Answer:
We agree with the respected reviewer that the proposed experiments (downregulation in normal cells, overexpression in SW780 in addition to function assays) will provide further information on the tumour suppressor/ promoter function of GSN, however due to logistic issues and time constraint the proposed work could not be achieved soon. Still awaiting the reagents which were ordered sometimes ago. The authors greatly appreciate and considering all the comments which will be published in a separate article at a later stage.

Reviewer 2 Report
1. Based on the reviewer’s experiences, measuring the p65 translocation in T24 cells is not difficult. The authors should observe the p65 translocation to prove NFkB activation.
2. In the discussion, the authors present the relationship between GSN and apoptosis in other cancers but not bladder cancer. That is why the authors should provide more data regarding the link of GSN to apoptosis in bladder cancer.
Some minor grammar errors are needed to be amended.
Author Response
The authors would like to thank the reviewer for the insightful and constructive comments, which help improving the quality of our manuscript. All responses to the comments are provided below in italics.
Reviewer#2
- Based on the reviewer’s experiences, measuring the p65 translocation in T24 cells is not difficult. The authors should observe the p65 translocation to prove NFkB activation.
Answer:
As per the reviewer’s recommendation, experiment measuring p65 translocation was performed and results were updated accordingly in the revised manuscript.
Immunofluorescence staining (IF) was performed following transfection of T24 cells with GSN-siRNA or control siRNA. The IF was performed using both GSN antibody to ensure the knockdown of GSN (Alexa fluor-555, red) and NF-kB antibody to monitor the nuclear translocation of NF-kB. Results are presented in new figure 6.
Reviewer#2
- In the discussion, the authors present the relationship between GSN and apoptosis in other cancers but not bladder cancer. That is why the authors should provide more data regarding the link of GSN to apoptosis in bladder cancer.
Answer:
As per the reviewer’s suggestion, we elaborated on the association between GSN and apoptosis in the discussion section.

Round 3
Reviewer 1 Report
I am really surprised that, even more than two months after the second revision, none of the issues raised have been addressed, even partially. The authors did not present any new data related to my comments, and, moreover, did not attempt to discuss my points in the manuscript, nor in their reply to me. That being the case, I am sorry, but it is quite obvious that I cannot recommend the publication of the article as it is.
Author Response
Dear Dr. Li,
We would like to thank you for giving us a chance to modify the manuscript in order to address the comments raised by the reviewers, and also thank the reviewers for their insightful and constructive comments, which help improving the quality of this manuscript. Our responses to the comments are attached. (The reviewer’s comments are in italics).

Reviewer 2 Report
no further comments
Author Response

(The authors gave the same response as above.)
